# The Relationship between Health Insurance and Pharmaceutical Innovation: An Empirical Study Based on Meta-Analysis

**DOI:** 10.3390/healthcare11222916

**Published:** 2023-11-07

**Authors:** Chenchen Fan, Xiaoting Song, Chunyan Li

**Affiliations:** Shanghai International College of Intellectual Property, Tongji University, Shanghai 200092, China; 2110990@tongji.edu.cn (C.F.); tcmip@126.com (X.S.)

**Keywords:** health insurance, pharmaceutical innovation, meta-analysis, effect size

## Abstract

The growing research interest in the relationship between health insurance and pharmaceutical innovation is driven by their significant impact on healthcare optimization and pharmaceutical development. The existing literature, however, lacks consensus on this relationship and provides no evidence of the magnitude of a correlation. In this context, this study employs meta-analysis to explore the extent to which health insurance affects pharmaceutical innovation. It analyzes 202 observations from 14 independent research samples, using the regression coefficient of health insurance on pharmaceutical innovation as the effect size. The results reveal that there is a strong positive correlation between health insurance and pharmaceutical innovation (r = 0.367, 95% CI = [0.294, 0.436]). Public health insurance exhibits a stronger promoting effect on pharmaceutical innovation than commercial health insurance. The relationship between health insurance and pharmaceutical innovation is moderated by the country of sample origin, data range, journal type, journal impact factor, type of health insurance, and research perspective. Our research findings further elucidate the relationship mechanism between health insurance and pharmaceutical innovation, providing a valuable reference for future explorations in pharmaceutical fields.

## 1. Introduction

As one of the crucial strategies for achieving social equality in the healthcare system, health insurance contributes to improving the accessibility of healthcare services and essential medicines, thus safeguarding the basic rights to life for residents and achieving universal health goals [1,2,3,4,5,6]. Health insurance exerts a direct influence on investment in the pharmaceutical industry and drives innovation by actively promoting the utilization of drugs and medical services [7,8,9,10]. Health insurance is vital for pharmaceutical innovation by ensuring access to necessary medications. Pharmaceutical innovation refers to the discovery and development of new drugs through scientific research, drug design, preclinical testing, and clinical trials [11]. It involves understanding the molecular mechanisms of diseases, identifying potential drug targets, and creating new chemical entities that can effectively treat, prevent, or manage various diseases. Pharmaceutical innovation helps to improve the overall efficiency of health insurance systems and population health outcomes by providing patients with access to more effective and safer treatment options and enables healthcare systems to deliver personalized and cost-effective care [12,13]. It reduces financial barriers and incentivizes pharmaceutical companies to invest in research and development. In effect, health insurance bridges the gap between affordability and medical breakthroughs, fostering a thriving market that encourages innovation.

While some have suggested that changes in health insurance coverage can have a positive impact on pharmaceutical innovation [14,15,16], empirical research has produced varied and even conflicting results, including positive correlation, negative correlation, and no correlation [17,18,19]. In general, there has not been a consensus in academic circles on the correlation between health insurance and pharmaceutical innovation. The mixed results of these studies might be due to the fact that diverse attributes have been investigated in the different studies. However, these attributes seem to have a rather different impact and might cause high variability and uncertainty in cases when a health insurance strategy is implemented.

Moreover, previous research only focused on whether health insurance impacts pharmaceutical innovation while neglecting the magnitude of the correlation between health insurance and pharmaceutical innovation. This leaves policymakers and other decision-makers with little direct evidence about the ultimate magnitude of the effect of policies on pharmaceutical innovation. Therefore, it is critical to understand the effect size of health insurance on pharmaceutical innovation if we are to design insurance policies in a manner that ensures the greatest expected health benefits for both current and future patients.

To fill the above research gaps, our objective is to contribute several new insights into the relationship between health insurance and pharmaceutical innovation. It aims to answer the following questions: First, does health insurance impact pharmaceutical innovation? Second, what is the magnitude of the correlation between health insurance on pharmaceutical innovation? Third, why do existing research results differ? A single empirical study may encounter issues with sampling, measurement, stochasticity, and external validity. However, a comprehensive and systematic analysis of the health insurance–pharmaceutical innovation relationship, to our best knowledge, has not been undertaken. Meta-analysis has a rigorous and systematic calculation procedure to aggregate prior empirical studies, so it can overcome the limitations in a single empirical study and address contentious conclusions [20,21]. It evaluates the heterogeneity, publication bias, and genuine effects of existing studies, thereby augmenting the value and scientific rigor of research and surmounting the limitations associated with qualitative methods. As a result, it provides more accurate and robust research conclusions, objectively explaining research differences to compensate for the deficiencies of narrative literature reviews. Additionally, it offers more valuable references for subsequent research [22].

Specifically, this study would identify a more objective presentation of the relationship and the magnitude of a correlation between health insurance and pharmaceutical innovation by systematically summarizing empirical research, conducting statistical re-analysis, and obtaining stage-specific conclusions with a low error and high reliability. Furthermore, we seek to explore the factors contributing to the variations in conclusions regarding the effects of health insurance on innovation. Various moderating factors, such as sample selection, variable characteristics, and literature features are investigated to comprehensively analyze the situations in which medical insurance has different effects on pharmaceutical innovation. The research findings of this study partially reflect the average impact of medical insurance on pharmaceutical innovation. Further to this, these findings make a valuable contribution to the development of conclusive and comprehensive scientific knowledge in this field, offering scientific evidence to support the formulation of health insurance policies. 

## 2. Hypotheses Development

### 2.1. The Impact of Health Insurance on Pharmaceutical Innovation

Scholars have extensively researched the impact of health insurance on pharmaceutical innovation. Among these studies, there is a large amount of empirical evidence that supports the promotion of pharmaceutical innovation by medical insurance [14,15,16,23,24]. Patent applications, R&D investment, and clinical trials are commonly used to measure pharmaceutical innovation. Patents remain the primary method of protecting innovation in the pharmaceutical industry compared to other research-intensive sectors [25]. Developing new drugs requires R&D investments and multiple clinical trials approvals. Prior study has shown that firms’ R&D investments increase upon implementation of Medicare Part D (prescription drug insurance) [26]. According to Blume-Kohout’s study, the implementation of Medicare Part D promotes drug innovation (measured by the number of clinical trials) through heightened drug utilization among the elderly population [16]. Opposing views have been proposed by some scholars. For instance, Peltzman believes that the strict government regulations and price controls on drugs covered by medical insurance policies have led to reduced efficiency, and prolonged the time for drug approval and market entry, thereby dampening companies’ enthusiasm for technological innovation [27]. In addition, critics argued that more generous health insurance coverage policies create perverse incentives for firms to develop expensive products with minimal incremental clinical value [28,29,30]. Despite the lack of consensus within the academic community regarding the relationship between health insurance and pharmaceutical innovation, strengthening health insurance coverage is an often-used lever through which policymakers can improve patients’ access to prescription drugs [31]. 

The demand-induced innovation theory believes that innovation is driven by market demand [32,33,34]. Potential market demand not only serves as a ‘direct incentive mechanism’ for innovation entities to engage in innovation but also indirectly incentivizes technological innovation by reducing the uncertainty of expected profits and operational risks. Therefore, based on the demand-induced innovation theory, this paper provides an explanatory framework for the impact of medical insurance on pharmaceutical innovation from several perspectives. 

At the macro level, expanding the scope of the medical insurance catalog to improve patient’s access to drugs is a common means employed by decision-makers [35]. The expansion of the coverage scope of medical insurance can induce innovation through the enlargement of the market size [34]. In addition, medical insurance policies indirectly drive pharmaceutical innovation by influencing the allocation of medical resources. Within the coverage scope of medical insurance, hospitals and patients are more inclined to use drugs listed in the medical insurance catalog, thereby affecting the demand for drugs. 

At the meso level, health insurance primarily influences pharmaceutical innovation by impacting the business strategies of pharmaceutical companies. From the perspective of pharmaceutical companies in the supply chain, the production of medicines is determined by the input and combination of production factors, while the sales of medicines are also determined by whether they are included in the medical insurance catalog and the prevalence of the disease. For pharmaceutical companies, drugs covered by the medical insurance catalog result in the demand and market size of the drugs expanding, increasing sales certainty and reducing the risk of uncertainty in the process of innovation. Thus, pharmaceutical companies reconfigure resource elements and promote innovation in the pharmaceutical industry to maximize profits [36]. At the same time, drugs within the medical insurance catalog have a higher market competitiveness. Enterprises will adjust their business strategies based on changes in the medical insurance catalog to enhance their market competitiveness. 

At the micro level, health insurance primarily influences pharmaceutical innovation by affecting patients’ medical consumption behavior and the quality of medical services. For patients, medical insurance can alleviate their burden of medical expenses, increasing their willingness and ability to consume medical services. This will promote the demand for new drugs by patients [37], thereby driving the innovation motivation of pharmaceutical companies. In addition, doctors, as providers of medical services and decision-makers for medication, play a crucial role in the relationship between medical insurance and pharmaceutical innovation. Based on their medical expertise and clinical experience, they formulate treatment plans for patients and determine the prescription and medication cycle. This decision-making process directly affects the category of drugs actually used by patients and the demand for drugs. Consequently, this study proposes the following hypotheses:

**H1:** 
*A positive correlation exists between health insurance and pharmaceutical innovation.*


Patents play a crucial role in measuring pharmaceutical technological innovation. Unlike new drug applications, which typically occur at the end of the drug discovery and development process, patent applications for new medicines can be filed at various stages of the innovation timeline. Generally, patent applications can be initiated as early as the preclinical testing stage of a new drug [25]. According to the theory of demand-induced innovation and the interaction between technology innovation and market demand, market demand greatly influences technological innovation [32,33,34]. The expansion of health insurance coverage increases the demand for pharmaceuticals for diseases covered by insurance, thereby stimulating innovation in the pharmaceutical industry. Expanded market demand incentivizes pharmaceutical enterprises to increase research and development investment, influencing technological innovation in the pharmaceutical industry. The broadening of health insurance coverage offers increased market opportunities and potential benefits for companies involved in pharmaceutical research. This prompts companies to actively engage in innovative research and development, pursue patents in new fields or markets to safeguard their innovations and investments and uphold their competitive advantage. This study proposes:

**H1a:** 
*A positive correlation exists between health insurance and patent applications.*


Investment in research and development predominantly gauges the level of enthusiasm and innovation capabilities exhibited by pharmaceutical innovation entities in technological advancements subsequent to health insurance coverage. Evidence indicates that the expansion of insurance coverage increased prescription drug use [38,39]. This increased use of prescription drugs resulting from the expansion of insurance coverage might yield increases in pharmaceutical firms’ R&D via two mechanisms. Firstly, prior studies have shown that firms’ R&D expenditures exhibit nearly unit elasticity in response to increases in sales revenues, which increased substantially upon implementation of Part D [40,41]. In addition, Duggan and Scott Morton found that pharmaceutical firms experienced an overall increase in their revenues after the implementation of Part D, despite the price decreases negotiated by private insurers [10]. Secondly, per Friedman’s research, stock prices for companies introducing high Medicare share drugs increased dramatically after the implementation of Part D, and increases in stock prices may decrease the cost of external capital, thereby increasing R&D expenditures [42,43]. This study proposes:

**H1b:** 
*A positive correlation exists between health insurance and R&D investment.*


The development of new drugs and therapies to combat serious and life-threatening diseases heavily depends on clinical trials. These studies evaluate the safety and effectiveness of new drugs and therapies being developed by involving human volunteers. Clinical trials are a crucial step in the drug innovation process and the development of new therapies for serious diseases. The absence of coverage for routine care costs was perceived by the scientific community and policymakers as a significant barrier to enrollment in clinical trials [44]. Health insurance reduces the financial burden on patients by covering the costs associated with trial participation, making it more accessible. Additionally, health insurance expands the patient pool by providing coverage, allowing for a more diverse and representative population to participate in trials. This, in turn, leads to more robust data and increased interest from researchers and sponsors. Moreover, the market’s demand for advanced healthcare options may encourage companies to increase clinical trials. By sorting out the original literature, the clinical trials in this study include the number of clinical trials and the number of drugs entering clinical trials. This study proposes:

**H1c:** 
*A positive correlation exists between health insurance and clinical trials.*


The theoretical analysis and research hypotheses presented above serve as the foundation for the ensuing study. This foundational framework is comprehensively delineated in Figure 1.

### 2.2. The Potential Moderating Variable

(1) Factors contributing to variations in the relationship between health insurance and pharmaceutical innovation at the sample level.

The source countries of the study samples are different. The health insurance policies and macro–micro environments in different countries vary, which can influence the relationship between health insurance and pharmaceutical innovation. According to several studies, public health insurance is the primary form of health insurance in China, and evidence suggests that a wider coverage of health insurance policies leads to a more significant impact on innovation. Zhang et al. conducted a natural experiment from the implementation of a public health insurance program for rural residents in China and found a 12.4% increase in relevant domestic pharmaceutical patent applications in diseases with a 10% higher number of rural patients [45]. This indicates that the implementation of public health insurance in China has positive spillover effects on pharmaceutical innovation, stimulating innovation in drugs for prevalent diseases in rural areas. Natalie Chun et al. conducted an analysis of the relationship between the implementation of health insurance policies in the United States and drug innovation [44]. The results showed that the policy did not lead to an increase in new drug clinical trials. Therefore, the source country of the sample enterprise may have an impact on the relationship between health insurance and pharmaceutical innovation. This paper proposes: 

**H2:** 
*Significant differences exist between China and the United States.*


(2) Factors contributing to variations in the relationship between health insurance and pharmaceutical innovation at the data level.

The data periods of the study samples are different. The study samples have varying time spans, which could potentially influence the relationship between health insurance and pharmaceutical innovation. Since the start of the 21st century, the coverage of medical insurance has progressively widened. In 2000, China released the first edition of the Basic Medical Insurance Drug List. The expansion of medical insurance coverage has changed the market demand for drugs listed in the corresponding catalogs, thereby influencing firms’ investment in innovation. Therefore, studies conducted during later data periods (before 2000) and earlier data periods (after 2000) may lead to different research findings. As an instance, Daron Acemoglu et al. conducted a study utilizing data spanning from 1965 to 1999, and their findings indicated the absence of a notable impact of health insurance on pharmaceutical innovation [46]. Conversely, Craig Garthwaite et al. conducted a study utilizing data spanning from 2004 to 2016 and identified a positive correlation between health insurance and pharmaceutical innovation [30]. Thus, the different time intervals of the sample data may affect the relationship between health insurance and pharmaceutical innovation. This paper proposes:

**H3:** 
*Significant differences exist in different periods of research data.*


(3) The sources of differences in the relationship between health insurance and pharmaceutical innovation at the literature level.

(a) The publication years of the literature are different. The publication focus of literature may differ across various years. In periods of high prevalence of medical insurance or pharmaceutical innovation research, there tends to be a greater inclination to publish literature demonstrating a significant positive association between health insurance and pharmaceutical innovation. On the one hand, in recent periods, the correlation between medical insurance and pharmaceutical innovation has emerged as a prominent research topic, capturing the interest of scholars. Therefore, researchers are more likely to emphasize the beneficial influence of health insurance on pharmaceutical innovation, aligning with academic trends and research priorities. On the other hand, governments and policymakers are typically dedicated to fostering the advancement of health insurance and pharmaceutical innovation. During a certain period of policy enactment, the biases existing in policies and the pervasive influence of the economic environment may prompt researchers to illustrate a positive association between health insurance and pharmaceutical innovation, aiming to bolster policy formulation and implementation. Consequently, variations in publication years can potentially affect the relationship between health insurance and pharmaceutical innovation. This study proposes:

**H4:** 
*Significant differences exist in different publication years of the literature.*


(b) The journal impact factors of the literature vary. Journals with different impact factors may focus on different research outcomes. Generally, studies demonstrating significant statistical results are more likely to be published. Journals with high impacts may prioritize the novelty of research findings and the significance of statistical outcomes to sustain their ongoing academic impact. Thus, journals with different impact factors can potentially lead to divergent patterns in the relationship between medical insurance and pharmaceutical innovation. This study proposes:

**H5:** 
*Significant differences exist between journals with high-impact factors and low-impact factors.*


(c) The types of literature are different. The literature selected in this study includes theses and journal papers. These are unpublished research, while journal papers are published research. Journal papers with statistically significant research results are more likely to be published than those without significance after undergoing peer review. Nevertheless, non-significant research results can offer more precise measurements of the actual correlation between variables. Thus, the various types of literature publications can lead to variations in the relationship between health insurance and pharmaceutical innovation. This study proposes:

**H6:** 
*Significant differences exist between theses and journal papers.*


(4) The sources of differences in the relationship between health insurance and pharmaceutical innovation at the variable level.

(a) The types of health insurance are different. By sorting out the original literature, the research samples include two types of health insurance: commercial and public. Commercial health insurance, known for its substantial funding and generous reimbursement, plays a crucial role in driving healthcare consumption and stimulating pharmaceutical innovation. The profitability of innovative drugs attracts increased capital investment in pharmaceutical research, fostering a virtuous cycle between market demand and ongoing pharmaceutical innovation. Leila Agha’s study illustrates that adjustments to commercial health insurance formularies significantly impact pharmaceutical innovation [47]. Pharmaceutical companies reduce investments in drugs facing high exclusion risks from these formularies. Public medical insurance has lower funding and inadequate reimbursement, limiting its ability to meet the health security needs of middle- and high-income groups, and reducing their purchasing power for innovative drugs. Therefore, the different types of medical insurance may result in different outcomes in the relationship between health insurance and pharmaceutical innovation. This study proposes:

**H7:** *Significant differences exist between commercial and public health insurance*.

(b) The studies employ diverse analytical perspectives. The research perspective of this article is divided into overall and partial perspectives based on the coverage of health insurance in the research samples. For example, Medicare in the United States includes four parts: A, B, C, and D. Some studies only focus on the Medicare Part D program, so this article classifies such studies into a partial analytical perspective. Some studies focus on all Medicare parts, which are classified into the overall analytical perspective in this article. Differences in analysis perspectives and measurement indicators can result in variations in the relationship between medical insurance and pharmaceutical innovation. For example, Blume-Kohout’s study revealed that the Medicare Part D program in the United States stimulates pharmaceutical innovation [48]. In contrast, Daron Acemoglu’s global research showed that the impact of medical insurance on pharmaceutical innovation is not substantial [46]. Therefore, the different analytical perspectives may have an impact on the relationship between health insurance and pharmaceutical innovation. This study proposes:

**H8:** 
*Significant differences exist in different analytical perspectives.*


## 3. Methodology Framework

This study employs the meta-analysis method developed by the American educationist Glass [49] in 1976 to investigate the influence of health insurance on pharmaceutical innovation. Meta-analysis utilizes specific criteria to statistically analyze and integrate findings from multiple empirical studies, enhancing study accuracy and providing valuable insights for future research. Compared to conventional quantitative studies, it exhibits heightened rigor and comprehensiveness in the process of acquiring, selecting, and evaluating original literature. The research methodology encompassed three primary stages. Initially, data collection from diverse databases was executed with meticulous adherence to predefined inclusion and exclusion criteria. Subsequently, the screened data were systematically categorized and encoded based on statistical and descriptive parameters. Lastly, the STATA software was employed for publication bias analysis, overall tests, and tests of the regulatory effect, culminating in the derivation of the ultimate research findings. Figure 2 visually outlines the research design employed in this study.

### 3.1. Literature Retrieval and Screening

A comprehensive and rigorous literature search was conducted to examine the empirical study exploring the influence of medical insurance on pharmaceutical innovation. Firstly, prominent databases, including Web of Science, PubMed, Scopus, PQDT, and CNKI, were scrutinized. Secondly, keywords such as “Insurance, Health”, “Health Insurances”, “Insurances, Health”, “Health Insurance”, “Health Insurance, Voluntary”, “Insurance, Voluntary Health”, “Voluntary Health Insurance”, “Group Health Insurance”, “Health Insurance, Group”, “Insurance, Group Health”, “Insurance, Group Health”, “medicare”, “Health care”, “Healthcare”, “Medical care insurance”, and “medical insurance” were combined with “pharmaceutical innovation” and “drug innovation” to search for literature with these keywords in the title or abstract. To avoid missing any documents, we carried out a manual supplementary search in the process of reading literature. With a deadline for literature retrieval settled as 15 June 2023, a total of 7138 literature records were collected through the operation mentioned above. 

By meticulously reviewing the titles, abstracts, and full texts of the acquired articles, any instances of duplication, irrelevance, and non-quantitative studies were gradually eliminated. In adherence to the selection criteria for meta-analysis, the distinct screening criteria applied in this study are elucidated as follows: (1) The research inquiry focused on the examination of the impact of medical insurance on innovation, imposing restrictions on the inclusion of literature that is irrelevant to the research topic or pertains to the development and operationalization of measurement scales. (2) The selected literature must consist of empirical studies that yield quantitative outcomes, providing a comprehensive disclosure of essential details such as sample size, variable reliability, correlation coefficients between medical insurance and various variables, or other effect values that can be transformed into correlation coefficients. Consequently, theoretical studies, qualitative research designs such as case studies, and literature reviews have been excluded. (3) It is essential to include literature that provides complete data information, encompassing statistical coefficients, sample size, *t*-values, and *p*-values. (4) The literature comprises independent research samples. If the literature contains multiple samples, all should be included in the database. If different kinds of literature use the same sample, only one is included. In cases of overlap, the literature with a larger sample size is chosen for analysis. Ultimately, 14 articles containing research samples from China and the United States were obtained in the meta-analysis database. The screening process of this study is shown in Figure 3.

### 3.2. Data Coding and Effect Size Calculation

Once the sample for analysis has been determined, it is essential to code and transform the original measurements extracted from the literature in order to calculate the key indicators for meta-analysis. This study adhered to the coding steps recommended by Lipsey et al. [50]. In order to guarantee coding data accuracy, two trained coders independently coded the aforementioned 14 articles. The coding data primarily consist of descriptive and statistical items. Descriptive items relate to research design and literature publication, encompassing authorship, title, publication date, keywords, and literature source. Statistical items cover sample size, data type, measurement indicators, journal type, journal impact factor, regression coefficients (such as β, t-values), and other variable characteristics. 

After coding the literature, key indicators for meta-analysis, namely effect sizes, can be calculated. Effect sizes are quantitative measurements of the strength of the relationship between two variables and the magnitude of differences between groups, representing practical significance. In research work, effect size generally refers to the correlation coefficient between variables. Drawing on the analysis methods of correlation coefficients or regression techniques commonly used in current management studies, this article also adopts the effect value of correlation coefficient (r-based) to represent the impact of health insurance on innovation. First, according to the formula proposed by Rosenthal [51], the estimated parameters (*t*-values) of the original study are converted into correlation coefficients (r). The calculation formula is r = $\sqrt{\left(t^{2}/(t^{2}+df)\right)}$. Here, df refers to the degrees of freedom associated with the t-value, which can be calculated based on the sample size and variable values in the original study. Due to differences in sample sizes among the original studies, it is necessary to correct for bias caused by sample differences. Therefore, the correlation coefficient should be converted into a standardized effect size called Fish-Z. The specific calculation method is as follows: (1) Calculate Zr (Zr = 0.5ln[(1 + r)/(1 − r)]); (2) Calculate the variance of z ($V_{Z}$/(n − 3)); (3) Calculate SE_z_ (SE_z_ = $\sqrt{V_{Z}})$. The calculations mentioned above were carried out using Stata 17 software. The basic unit of effect size is an independent sample. If multiple independent samples appear in the original studies, multiple coding is required. The final consistency coefficient of the coding was 93% (>90%), reaching a reliable level, indicating that the coding in this study was effective [52]. Following the completion of the coding process, a total of 202 valid effect sizes were obtained from a set of 14 empirical studies, with 152 effect sizes exceeding 0 and 50 effect sizes below 0.

## 4. Results of the Meta-Analysis

### 4.1. Analysis of Publication Bias

Meta-analysis relies on published literature, which may not fully represent the overall research in the field as it tends to favor studies with significant results. To ensure the reliability and accuracy of the meta-analysis results, this study investigated publication bias within the literature. Common methods used to assess publication bias include the fail-safe coefficient method, funnel plot method, Begg’s test, the trim-and-fill method, and Egger’s test [53]. This study used two methods to investigate the presence of publication bias in the results across various dimensions. 

As shown in Figure 4, this study displayed the sample literature analyzed using the funnel plot. The findings demonstrated evenly distributed effect sizes surrounding the overall effect size, with the funnel plot showing an inverted shape, indicating no publication bias. To enhance reliability, this study used the fail-safe coefficient method to investigate publication bias further.

The fail-safe coefficient method is a quantitative approach employed for the evaluation of publication bias. It determines the number of unpublished studies with null results that would be necessary to reduce the cumulative effect to a statistically insignificant level. As shown in Table 1, the fail-safe coefficient in this study was 85,849, significantly higher than 1005, indicating robust and unbiased results.

### 4.2. Overall Test

This study conducted an overall test on the included effect sizes and their standard errors, which consists of a heterogeneity test and model result evaluation. By combining the effect sizes, the reliability of the hypothesis regarding the relationship between health insurance and pharmaceutical innovation was comprehensively assessed. The heterogeneity level is typically assessed using the Q statistic, its significance level, and I^2^ [54]. When Q > df(Q), *p* < 0.05, and I^2^ > 75%, the results of various studies are considered heterogeneous, and a random-effects model should be chosen to combine the effect sizes. Conversely, a fixed-effects model should be selected.

Table 2 presents the results of the overall test for health insurance and pharmaceutical innovation. From the perspective of the heterogeneity test, Q = 3213.01 > 198, *p* < 0.05, and I^2^ = 93.84%, indicating a high heterogeneity among the 202 effect sizes included in the meta-analysis. The true differences in effect sizes account for 93.84% of the observed variability, while the remaining 6.16% is due to random error. Therefore, a random-effects model should be selected. The variance value is 0.324, indicating that 32% of the variation between studies can be used to calculate the weights.

The results of the model testing show a correlation coefficient of 0.367 between health insurance and pharmaceutical innovation, with a 95% confidence interval of [0.294, 0.436]. Gignac and Szodorai provided guidelines for the purposes of interpreting the magnitude of a correlation [55]. Specifically, r = 0.10, r = 0.20, and r = 0.50 were recommended to be considered small, medium, and large in magnitude, respectively. Based on this standard, the correlation coefficient between health insurance and pharmaceutical innovation in this study exceeds 0.3, indicating a robust positive correlation and confirming Hypothesis 1. The correlation coefficients for the pharmaceutical patent applications, pharmaceutical research and development investment, and clinical trials are 0.485, 0.159, and 0.258, respectively. All coefficients are statistically significant within the 95% confidence interval (*p* < 0.05). This suggests that health insurance has a positive impact on the number of pharmaceutical patent applications, R&D investment, and clinical trials, thus validating hypotheses H1a, H1b, and H1c. Among them, patent applications and R&D investment are highly positively correlated with medical insurance, while the number of clinical trials is moderately positively correlated with pharmaceutical innovation.

Meta-analysis scientifically evaluates the relationship between health insurance and pharmaceutical innovation as a whole, addressing the question of whether health insurance promotes pharmaceutical innovation. Furthermore, we found that there exists a large positive correlation between health insurance and pharmaceutical innovation. However, significant variations exist in the effect sizes across different research reports. Possible moderating factors may influence the strength of the effect size, necessitating further investigation to identify reasons for the inconsistency in effect sizes.

### 4.3. Moderation Effect Testing

The previous section’s analysis of overall effect testing revealed heterogeneity in this study, suggesting that the role of health insurance in pharmaceutical innovation is impacted by potential moderating variables. In order to test the reliability of the research conclusions and analyze the reasons for the inconsistency of the conclusions, subgroup analysis needs to be used to test the moderating variables [56]. Table 3 reports the results of the subgroup analysis. Specifically: (1) The P values of the moderation tests for the United States and China (*p* = 0.001 < 0.05) both reached significance, indicating that the country of sample origin has a significant moderating effect on the relationship between health insurance and pharmaceutical innovation. Health insurance policies have a stronger promoting effect on pharmaceutical innovation in the Chinese sample (r = 0.473) compared to the US sample (r = 0.236). Thus, Hypothesis H2 is validated. (2) The moderation test yielded a significant *p* value (*p* = 0.001 < 0.05), indicating that the data’s starting year has a stronger promoting effect on pharmaceutical innovation before the year 2000 (r = 0.433) compared to after the 2000 year (r = 0.173). Therefore, Hypothesis H3 is supported. (3) The moderation test yielded a significant P value for publication years (*p* = 0.000 < 0.05), suggesting that the publication year has a significant moderating effect on the relationship between health insurance and pharmaceutical innovation. The promoting effect of health insurance on pharmaceutical innovation is stronger before 2010 (r = 0.367) compared to after 2010 (r = 0.179). Therefore, Hypothesis H4 is validated. (4) The relationship between health insurance and pharmaceutical innovation is significantly moderated by the journal impact factor, as evidenced by a Q value (between groups) of 2535.24 and *p* = 0.034. These findings indicate that a higher journal impact factor is associated with more positive results in the relationship between health insurance and pharmaceutical innovation. Thus, Hypothesis H5 is supported. (5) The moderation effect of document type on the relationship between health insurance and pharmaceutical innovation is not significant. Therefore, Hypothesis H6 is not validated. (6) Health insurance type (*p* = 0.014 < 0.05) can significantly moderate the relationship between health insurance and pharmaceutical innovation. Therefore, Hypothesis H7 is validated. (7) The relationship between health insurance and pharmaceutical innovation is significantly moderated by the analytical perspective, as confirmed by a Q value (between groups) of 37.63 and *p* = 0.000. These findings suggest that a broader research perspective is associated with more positive results in the relationship between health insurance and pharmaceutical innovation. Thus, Hypothesis H8 is validated.

## 5. Discussion and Conclusions

### 5.1. Summary and Conclusion

Prior perspectives and findings on the relationship between health insurance and pharmaceutical innovation have shown inconsistency, yet no research has emerged to clarify this issue. This study conducted a meta-analysis to assess the overall relationship between health insurance and pharmaceutical innovation, revealing a strongly positive correlation. This finding demonstrates that health insurance positively influences pharmaceutical innovation, supporting the initial perspective and resolving the debate regarding the relationship’s magnitude and direction. This study refutes the notion of a negative or non-existent correlation between health insurance and pharmaceutical innovation, highlighting the statistically significant relationship that should not be disregarded or overstated in practical applications. 

Specifically, during the process of promoting pharmaceutical innovation, health insurance has a positive impact on pharmaceutical patent applications, R&D investment, and clinical trials. Firstly, health insurance can encourage pharmaceutical companies and research institutions to conduct new drug research and increase the number of patent applications. Health insurance plays a significant role in alleviating the economic burden on patients during treatment by providing reimbursement or compensation for medical expenses. This, in turn, promotes consumption upgrading within the pharmaceutical industry. As a result, there is an increased market demand for new drugs, which incentivizes pharmaceutical companies to engage in research and development activities and file more patent applications to safeguard their innovative achievements. The popularity of health insurance serves as an economic stimulant and motivator for pharmaceutical innovation. When drugs are included in the health insurance catalog, enterprises receive a substantial number of purchase orders, enabling them to swiftly establish dominance in the pharmaceutical market and achieve significant profits. In addition, the inclusion of drugs in the health insurance catalog signals superior drug quality, mitigating sales risks and reducing promotion costs for enterprises. This, in turn, enables them to allocate greater funds towards pharmaceutical research and development. Consequently, pharmaceutical companies and research institutions are encouraged to increase their investment in research and development, including securing additional research funding, acquiring laboratory equipment, and expanding their research team. These investments expedite the pace of drug innovation, enhancing the quality and quantity of new drug development. Furthermore, health insurance facilitates the execution of clinical trials by enabling a larger pool of patients to participate, thereby generating a richer dataset and an increased number of cases for drug development. Moreover, health insurance offers economic assistance to clinical trials, alleviating the financial burden on participants and facilitating trial conduction, as well as the recruitment of a larger number of patients. This plays a critical role in substantiating and verifying the safety and effectiveness of new drugs. 

### 5.2. Discussion

The overall conclusions drawn from the meta-analysis do not negate specific studies that have not been supported in actuality. Here, only the simple correlation between two variables is considered, and the degree of closeness of their relationship is likely to be influenced or interfered with by other variables. This study found that: 

Firstly, at the sample level, the country of sample origin has a significant moderating effect on the relationship between health insurance and pharmaceutical innovation. Compared to the United States, studies with China as the sample showed a stronger promotion of pharmaceutical innovation by health insurance. This indicates that despite the later start of China’s health insurance system, the Chinese government has made consistent improvements and developments through a series of reform measures, leading to broader coverage. According to the 2018 World Health Statistics report published by the World Health Organization (WHO) [57], China has successfully achieved universal basic health insurance coverage, encompassing a population of 1.35 billion and achieving a coverage rate exceeding 95%. In contrast, the United States lacks a universal health insurance system, resulting in a significant number of uninsured or underinsured individuals. Therefore, health insurance has a more pronounced promoting effect on pharmaceutical innovation in China.

Secondly, at the data level, various data intervals significantly moderate the relationship between health insurance and pharmaceutical innovation. In recent data intervals, there has been higher economic development, improved living standards, increased disposable income, rising public health awareness, and continuous enhancement of residents’ capacity and willingness to consume medical services. With the rise in chronic diseases, expanding health insurance coverage, increasing life expectancy, and an aging population, the demand for pharmaceuticals among residents has grown consistently. These factors have become the most important driving force for the innovative development of the pharmaceutical industry in China. Therefore, compared to data from earlier intervals, recent data from closer intervals in research show a stronger impact of health insurance on pharmaceutical innovation. 

Thirdly, at the literature level, the publication year and impact factors exhibit a significant moderating effect on the relationship between health insurance and pharmaceutical innovation, while the type of journal does not. Recent studies have demonstrated a stronger impact of health insurance on pharmaceutical innovation compared to earlier research, highlighting the increasingly vital role of health insurance policies in supporting and ensuring the sustainable development of pharmaceutical innovation. In comparison to journals with low impact factors, publications in journals with high impact factors have a stronger promotion effect of health insurance on pharmaceutical innovation. This suggests that journals with higher impact factors place more emphasis on the significance of statistical results. In addition, this study found that the moderating effect of journal type on the relationship between health insurance and pharmaceutical innovation is not significant. Despite journal articles being subject to peer review and tending to publish positive evaluations of policy benefits associated with a positive correlation between the two, there is no significant difference in overall publication patterns, indicating the stability of the relationship between health insurance and pharmaceutical innovation across journal types.

Fourthly, at the variable level, the type of health insurance and research perspective significantly moderate the relationship between health insurance and pharmaceutical innovation. In terms of health insurance type, the study sample includes both commercial health insurance and public health insurance. Despite its strong role in promoting the upgrading of medical consumption with its characteristics of “high fundraising and high payout,” commercial health insurance faces challenges due to its limited audience and difficulty in attracting larger market demand. Conversely, while the level of medical consumption may be low for the recipients of public health insurance, its coverage is more extensive, resulting in a greater demand for medicines. As a result, public health insurance exhibits a stronger promoting effect on pharmaceutical innovation. Regarding health insurance, China primarily offers urban, rural, and resident medical insurance, while the United States provides Medicare parts A, B, C, and D. Some papers adopt a partial perspective by focusing on specific types of health insurance, while others take a holistic approach in their studies. The holistic perspective provides insights into the overall development trajectory of health insurance; however, it fails to capture the distinctions among different types of health insurance. Consequently, this limitation may undermine the overall influence of health insurance on pharmaceutical innovation. Therefore, studies focusing on a partial perspective can effectively leverage their localized advantages and maximize the influence of health insurance on pharmaceutical innovation. In a word, the results of various variables are summarized in Table 4.

### 5.3. Limitations and Future Research Direction

This study systematically analyzes the existing literature, elucidating the research concerning the relationship between health insurance and pharmaceutical innovation. Furthermore, it provides empirical evidence to affirm this relationship, while addressing the limitations in variable selection and measurement that afflicted prior isolated studies. As a result, the study arrives at a more comprehensive and unbiased research conclusion. Nevertheless, a few limitations persist in this investigation. Firstly, common with most meta-analyses, our study is susceptible to methodological limitations [58,59]. Despite our diligent efforts to comprehensively search for pertinent literature, we encountered challenges in incorporating all studies, particularly those that remained unpublished. Additionally, only empirical examination references were selected, and documents that involved qualitative analyses such as case studies were not included in the meta-analysis. This circumstance may result in an insufficiency of information, impeding a thorough and comprehensive exploration of the relationship between the two. And, the meta-analysis relies on existing studies, and as a result, the conclusions drawn do not offer novel research perspectives. Hence, it has the potential to restrict our comprehension of the research.

Secondly, there are still significant variations in the relationship between health insurance and pharmaceutical innovation, indicating that there are many other factors influencing the relationship that have not been fully explored. For example, the therapeutic effect of drugs and the relationship between drugs and medical services will also influence the research results [60]. Drugs with poor therapeutic effects not only fail to meet treatment needs but also carry a higher risk of being excluded from the medical insurance catalog, thus affecting the relationship between the two. Medical services encompass a series of services required for treating diseases, including diagnosis, treatment, and care. In the process of medical services, the professional competence of doctors will affect the demand for and management of drugs, thereby influencing the results of the relationship between the two. Limited by the original literature of meta-analysis, the above moderating variables cannot be obtained, potentially simplifying the relationship between health insurance and pharmaceutical innovation.

Future research has the potential to address our current limitations in the following aspects: (1) Future research can endeavor to employ novel methodologies in examining the relationship between health insurance and pharmaceutical innovation, thereby surpassing the limitations inherent in meta-analysis approaches. For instance, employing a qualitative comparative analysis (QCA) allows for the incorporation of a broader range of non-quantitative literature in the analysis process, leading to more precise research outcomes. (2) Future research can delve deeper into other potential moderating variables, thereby further enriching the relevant theoretical model. We recommend that future research on this topic take into account the internal distinctions within medical insurance. For instance, examining factors like the doctor–patient relationship and the treatment provided in medical services may unveil significant variations in their correlation. (3) There may yet exist mediating variables within the relationship between the two. Subsequent research may enhance the comprehension of this relationship through the incorporation of fresh variables and theoretical frameworks.

## Figures and Tables

**Figure 1 healthcare-11-02916-f001:**
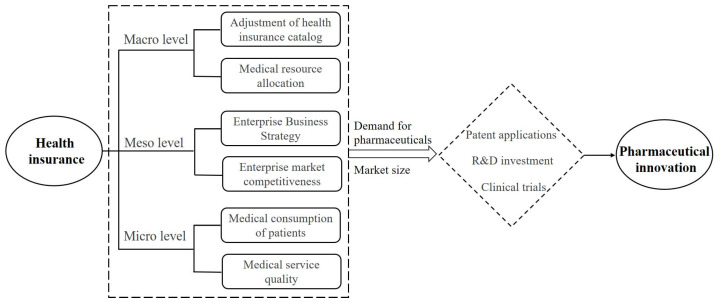
Meta-analysis diagram of the relationship between health insurance and pharmaceutical innovation.

**Figure 2 healthcare-11-02916-f002:**
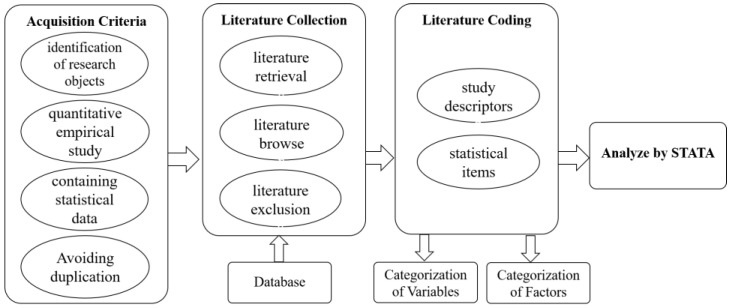
Flowchart of meta-analysis of the relationship between health insurance and pharmaceutical innovation.

**Figure 3 healthcare-11-02916-f003:**
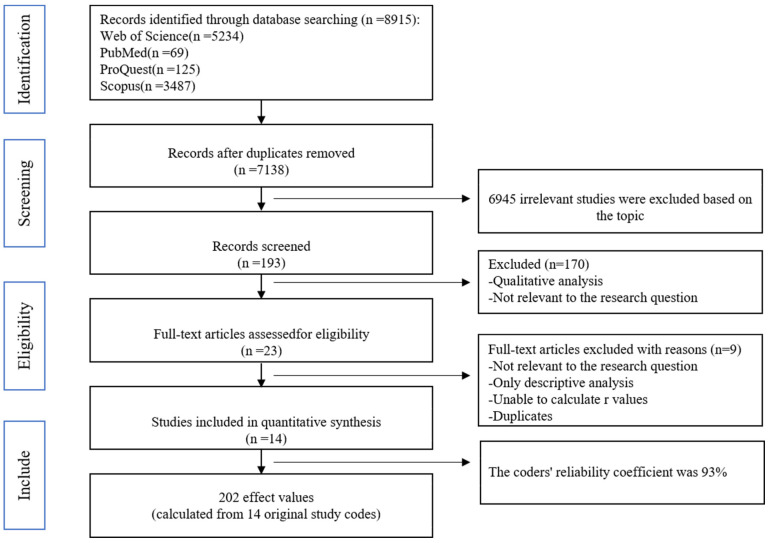
Flowchart of literature search, selection, and effect size coding.

**Figure 4 healthcare-11-02916-f004:**
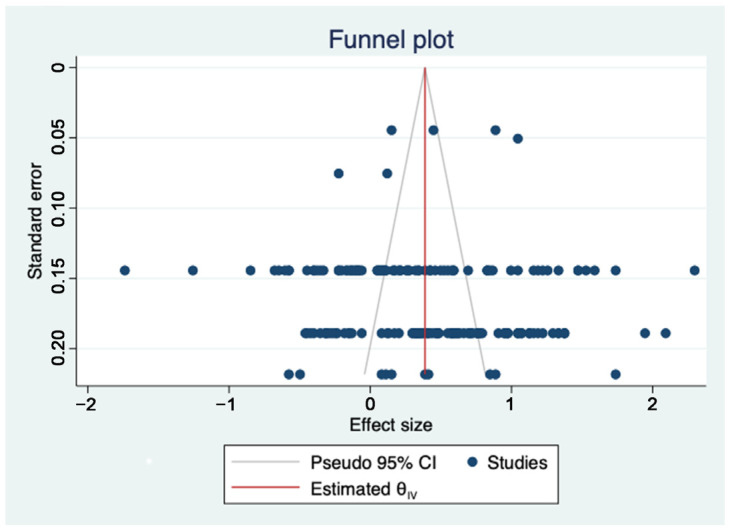
Funnel diagram of publication bias.

**Table 1 healthcare-11-02916-t001:** Test of publication bias.

Category	Sample Size	Fail-Safe Number	
	*K*	Threshold	*N_fs0.05_*
All	199	1005	85,849
Patent application	103	525	35,575
R&D investment	5	35	143
Clinical trials	75	385	6922

**Table 2 healthcare-11-02916-t002:** Meta results of the health insurance and pharmaceutical innovation.

Variable	Heterogeneity Test				Random Effects Model					The Magnitude of A Correlation
	*Df*	*p*	*I*^2^(%)	*Q*	*z*	Variance	Point Estimation	Lower Limit	Upper Limit	
Overall	198	0.000	93.84	3213.01	9.15	0.324	0.367	0.294	0.436	large
Patent application	102	0.000	90.05	1025.12	10.64	0.220	0.485	0.407	0.556	large
R&D investment	4	0.000	97.72	175.29	0.53	0.438	0.159	−0.408	0.637	small
Clinical trials	74	0.000	95.13	1520.22	3.49	0.407	0.258	0.115	0.390	medium

**Table 3 healthcare-11-02916-t003:** Meta results of the moderating effects.

Variables	Category	*k*	95%CI			Heterogeneity Test			
			Estimated Value	Lower Limit	Upper Limit	Q	Df	*p*	I^2^(%)
Country	US	92	0.236	0.108	0.357	10.65	198	0.001	93.84
	China	107	0.473	0.394	0.545				
Period	<2000	147	0.433	0.339	0.518	11.33	198	0.001	93.84
	≥2000	52	0.173	0.048	0.293				
Publication year	<2010	23	0.179	−0.222	0.529	3213.01	198	0.000	93.84
	≥2010	176	0.367	0.294	0.436				
Impact factor	<5	112	0.442	0.363	0.516	2535.24	172	0.034	93.22
	≥5	62	0.247	0.070	0.409				
Document type	Journal	173	0.375	0.292	0.453	0.02	188	0.891	93.80
	Thesis	16	0.359	0.132	0.551				
Health insurance type	Commercial insurance	3	−0.188	−0.571	0.262	5.99	190	0.014	94.00
	Public insurance	188	0.374	0.297	0.445				
Research perspective	Partial	165	0.431	0.354	0.502	37.63	190	0.000	94.00
	Overall	26	−0.098	−0.247	0.056				

**Table 4 healthcare-11-02916-t004:** Meta results of various variables.

Variables	Category	Subdimension	Hypothesis	Significance	The Magnitude of a Correlation
				Yes/No	
Core variables and the subdimensions	Overall	/	H1	yes	large
	/	Patent application	H1a	yes	large
	/	R&D investment	H1b	yes	small
	/	Clinical trials	H1c	yes	medium
Moderating variables and the subdimensions	Country	US	H2	yes	medium
		China			large
	Period	<2000	H3	yes	large
		≥2000			small
	Publication year	<2010	H4	yes	small
		≥2010			large
	Impact factor	<5	H5	yes	large
		≥5			medium
	Document type	Journal	H6	no	/
		Thesis			/
	Health insurance type	Commercial insurance	H7	yes	small
		Public insurance			large
	Research perspective	Partial	H8	yes	large
		Overall			small

## Data Availability

The Stata codes for this study can be provided by the author under request. No new data were generated. All data resources are publicly available.

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
