# Peer review of "The Relationship between Health Insurance and Pharmaceutical Innovation: An Empirical Study Based on Meta-Analysis"

_healthcare, 2023, doi:10.3390/healthcare11222916_

Round 1

Reviewer 1 Report

Comments and Suggestions for Authors

Summary 

This manuscript presents a meta analysis to explore how health insurance affects pharma innovation. The authors find a strong positive correlation between health insurance and pharma innovation and that the relationship is moderated by a number of factors.  

General Comments

This manuscript presents research that contributes to the literature of health insurance and pharmaceutical innovation.  The research appears to have been rigorously executed and arrives at meaningful conclusions.  However, there are a few areas where the paper could be strengthened. 

Pharmaceutical innovation is an imprecise concept and needs to be defined earlier in the paper.  Not until the hypotheses are introduced (page 3) does it become clear that innovation means patent applications, R&D, and the number of clinical trials.  

The abstract ends with the statement: “Our research findings are meaningful for policy makers to choose an appropriate healthcare strategy according to their unique attributes, enabling sustainable pharmaceutical development.”  This statement is puzzling because governmental policy is not discussed in the findings or conclusion sections of the paper. Moreover, the term “healthcare strategy” is unclear.  What are the strategic choices available to governments?  The authors should consider removing this statement.  

Hypotheses H1, H1a, H1b, and H1c are stated in an unambiguous manner (e.g., a positive correlation exists….).  The other hypotheses include the word “may” and thus are stated in an ambiguous manner (e.g., significant differences may exist). I suggest rewriting all hypotheses in the same manner as H1, H1a, H1b, and H1c. 

Research studies based on the US and China are emphasized in the manuscript.  Did the analysis cover the health insurance industry in any other country, or was the analysis restricted to these two countries?

Additional Comments

Ln 54-55.  The first question posed is:  “what is the impact of health insurance on pharmaceutical innovation?”  A better expression of the question might be: “does health insurance impact pharmaceutical innovation“ or “what is the directional impact of health insurance on pharmaceutical innovation?”  

Ln 154. Should “research and R&D” simply be “R&D”?

Ln 169.  H1c. posits a relationship between health insurance and clinical trials.  How are clinical trials defined in the studies used for the meta analysis? Number of clinical trials conducted? Number of participants?  Length of clinical trials? 

Ln 210-229.  This paragraph introduces research methods as a moderating variable. Different modeling and testing approaches surely explain differences in findings.  Dif-in-dif, GMM, and Poisson regressions are examples of the methods employed in the literature.  It seems simplistic to reduce the approaches to two: regression models and others.  This might explain why this moderating factor was not found to be important.  I recommend removing H4 from the paper. If H4 remains in the paper, then a more extensive discussion of the research methods used in this literature is needed.  

Ln 287-288.  H8 posits differences across different types of health insurance.  Would a more precise statement of H8 be that differences exist between commercial and public health insurance?  Or does “types of health insurance” include more than public and commercial health insurance? 

Ln 289-299.  H9 focuses on different analytical perspectives.  The setup for this hypothesis is confusing. I do not understand what is meant by analytical perspectives.  What are the categories of analytical perspectives used to conduct this test?

Ln 396.  Table 1 presents the sample size for the categories of innovation.  It would be useful to present these sample sizes in the text of the paper, especially given the different sizes for patent applications versus R&D investments.

Reviewer 2 Report

Comments and Suggestions for Authors

Presented reserarch takes focus on investigating the relationship between  Health Insurance and Pharmaceutical Innovation. Existing studies are reviewed, existing literature is reviewed, to derive hypotheses. Meta-analysis is applied to investigate the relationship emporically and to validate presented hypotheses. Overall the paper is difficult to read from my perception due to a missing illustration of assumend cause-effect relationships. In addition, I found the focus on this particular relationship to simplified, as medication and medical services (e.g. medical doctors) are not reflected. Furthermore, outcome and results of medications are not assessed. Implicit dependency of market creating measures (insurance finances drugs and medication), demand increases (however, the reason is not further explored), response by suppliers through increase of innovation. The concept innovation is not further explained and defined. Actors like patients and there agency is not considered. I would have expected rather a multi-level analysis framework (e.g. macro, meso, micro), as reality appears to be much more complex as the simplified model. In particular, no evaluation of the results is foreseen. The nature and characteristics of processed data remains as well vague for the reader. Selection of literature is not sufficiently well described. Empirical measures and methods are the stronger part of the paper but can be as well further elaborated, why the meta-analysis is most appropriate. The argumentation in general was not convincing in terms why this relationship is of relevance for policy makers, as important actors with agency are not considered and reflected in the research design, e.g. patient medical doctor relationship. In conclusion, the manuscripts has significant flaws and shortcomings and from my point of view is not publishable in its current form.  

Comments on the Quality of English Language

English is fine, please check for repitions as especially "relationship between  Health Insurance and Pharmaceutical Innovation" is redundant and repeatet to often limiting overall  readability.

Reviewer 3 Report

Comments and Suggestions for Authors

Fan et al. explored the impact of health insurance on pharmaceutical innovation by conducting a meta-analysis of observations from independent research samples. They report a strong positive correlation between health insurance and pharmaceutical innovation as well as the variables that affect this relationship. The study may appeal to some parties including government agencies, health insurance providers, investors, and policy makers. However, I would like the authors to address the following comments before this manuscript can be endorsed for publication.

Please mention in the methods section if any softwares or online servers have been used to facilitate the data process and/or analysis.

I suggest the authors address the effect of any confounding variables in these studies that may influence their conclusion and probably discuss these in the limitations section of the manuscript.

Round 2

Reviewer 2 Report

Comments and Suggestions for Authors

The authors have well responded to reviewers content. Submitted revision of the manuscript has considerably improved.  The overall approach and model is now better explicated and visible. The authors have added additional text and have improved the underlying conceptual framework. My recommendation is to insert a table showing all hyotheses being tested and if they are supported or not validated. Section "Discussion and Conclusion" needs to be separated into "Dicussion" and "Summary and Conclusion"; finally I would add a section "Limitations" to highlight some of the shortcomings and limitations of the meta-analysis, as well to indicated, how the results need to used or which issues need to be addressed by follow up activities. Recommendations for future research and open issues need to be added to the conclusion section. All other parts now look fine.

Comments on the Quality of English Language

English language is appropriate and the manuscript reads fine. However, languate needs to be finally checked, no significant shortcomings are visible.
